# Diseases Caused by Amoebae in Fish: An Overview

**DOI:** 10.3390/ani11040991

**Published:** 2021-04-01

**Authors:** Francesc Padrós, Maria Constenla

**Affiliations:** Departament de Biologia Animal, de Biologia Vegetal i d’Ecologia and Servei de Diagnòstic Patològic en Peixos, Universitat Autònoma de Barcelona, 08193 Barcelona, Spain

**Keywords:** *Neoparamoeba perurans*, AGD, NGD, systemic amoebiasis, *Endolimax nana*

## Abstract

**Simple Summary:**

Amoebae can be found in many different aquatic environments and are also an emerging risk for fish health. Amoebae can display different types of relationships with fish, some of them (amoeba acting as commensals) do not harm fish. However, in many cases they can act as parasites and can be the cause of severe diseases affecting mainly the gills and also causing relevant systemic infections.

**Abstract:**

Parasitic and amphizoic amoebae are ubiquitous and can affect a huge variety of hosts, from invertebrates to humans, and fish are not an exception. Most of the relationships between amoebae and fish are based on four different types: ectocommensals, ectoparasites, endocommensals and endoparasites, although the lines between them are not always clear. As ectocommensals, they are located specially on the gills and particularly the amphizoic *Neoparamoeba perurans* is the most relevant species, being a real pathogenic parasite in farmed salmon. It causes amoebic gill disease, which causes a progressive hyperplasia of epithelial cells in the gill filaments and lamellae. Nodular gill disease is its analogue in freshwater fish but the causative agent is still not clear, although several amoebae have been identified associated to the lesions. Other species have been described in different fish species, affecting not only gills but also other organs, even internal ones. In some cases, species of the genera *Naegleria* or *Acanthamoeba*, which also contain pathogenic species affecting humans, are usually described affecting freshwater fish species. As endocommensals, *Entamoebae* species have been described in the digestive tract of freshwater and marine fish species, but *Endolimax nana* can reach other organs and cause systemic infections in farmed *Solea senegalensis*. Other systemic infections caused by amoebae are usually described in wild fish, although in most cases these are isolated cases without clinical signs or significance.

## 1. Introduction

Generally speaking, the term amoeba describes a particular type of unicellular organism characterised by the ability to modify its shape mainly by the development of pseudopods. In the past, most amoebae were placed in the single Rhizopoda clade of the Sarcodina supergroup. However, recent studies based on molecular biology clearly indicate that amoebae are no longer a single taxonomic group but represent a clear example of a polyphyletic group, including members of different Supergroups, including Amoebozoa, Rhizaria, Excavata, Heterokonta, Alveolata and Opistokonta.

Amoebae, due to their characteristics, live in aquatic environments or can be present in aqueous fluids and humid environments and they can be also associated with external surfaces and the internal environments of animals. For this reason, most of the amoebae can be found mainly as free-living organisms (FLA) in aquatic environments [1], including natural environments but also man-made water storage and networks, such as drinking water or industrial cooling towers, and are also in soils. In addition, some species can also live in association with other organisms as ecto and endocommensals and also as ecto and endoparasites, acting in some of these cases as parasites and in many cases can alternate FLA phases with phases with a different level of association with organisms.

## 2. Diseases Caused by Amoebae in Human and Veterinary Medicine

As pathogens, amoebae are well-known organisms in human and veterinary medicine. In human medicine, several species have been described as pathogens or opportunistic pathogens. Amongst these species, probably the most well-known species and frequently reported in clinical cases in humans are the free-living amphyzoic amoeba *Acanthamoeba* spp., *Naegleria fowleri*, *Balamuthia mandrillaris* and *Sappinia* sp. [2]; and the *Entamoeba* complex [3,4] (particularly *E. histolytica* but also other opportunistic species such as *E. coli*), but also *Endolimax nana* or *Iodamoeba buetschlii* as the most pure form of endocommensals.

Amphyzoic amoebae, unlike true parasites, are not well adapted to parasitism, so they tend to be very aggressive within the host, causing their death in most cases. For instance, *Acanthamoeba* spp. is one of the most commonly isolated amoebae in environmental samples and they can cause granulomatous amoebic encephalitis (GAE), cutaneous acanthamoebiasis and amoebic keratitis (AK) in humans; or the percolozoan *Naegleria fowleri* causes primary amoebic meningoencephalitis (PAM) in humans, which is an acute, fulminant and fatal disease [5]. Water supplies, swimming pools, freshwater ponds, lakes and thermally polluted waters have been recognised as sources of human infections.

As endocommensal or endoparasite species of the human intestine *Entamoeba histolytica* is the only species that is considered pathogenic. However, *Entamoeba dispar, E. moshkovskii*, *E. hartmanni*, *E. coli*, *E. polecki*, *Endolimax nana* or *Iodamoeba buetschlii* are also very frequent and they are usually considered non-pathogenic. Infections by *Entamoeba histolytica* usually include an intestinal phase that ranges from asymptomatic colonisation to severe invasive infections (dysentery, colitis); and an extraintestinal phase, generally affecting the liver (amoebic liver abscess), with eventual progression to other organs (lung, brain, heart) through blood dissemination [6]. Apart from *E. histolytica,* other archamoebae species are able to cause diseases in their hosts. For example, gastrointestinal disorders have been reported associated with diverse species of *Entamoeba* [7,8], but also by *Endolimax nana* [9]. Furthermore, *E. dispar* has also been reported causing hepatic lesions [10] and *E. nana* has been described to be involved in skin processes and rheumatoid arthritis (see [11] and references herein). *Iodamoeba buetschlii* has also been reported to be able to cause brain granuloma [12].

In veterinary medicine, amphyzoic amoebae have also been reported in different vertebrates, such as non-human primates, dogs, bulls, horses, sheep, kangaroos… [13] and their pathogenesis varies depending on the species involved. They generally cause similar lesions to those in humans; granulomatous inflammatory lesions in nervous tissue, but also lesions in other organs or even systemic infections. Similarly, endoparasitic amoeba are also described in other vertebrates with different degrees of pathogenicity. *Entamoeba nuttalli or E. invadens* are highly related species *to E. histolytica* with similar pathogenicity, that can affect non-human primates and reptilians (see [14] and reference herein), or *E. bovis* and *E. ovis* infecting ruminants [15,16]. *Iodamoeba* and *Endolimax* are also endocommensal or facultative parasitic organisms in the intestinal tract of other vertebrates. Species of *Endolimax* have been reported for a variety of vertebrate hosts [17,18,19,20,21,22,23]. Although most of them are considered endocommensals, some of them are pathogenic.

In addition to the effect of primary pathogens in human and animal diseases, it is also very important to remark the recently described role of amoeba as reservoirs for human and animal pathogenic bacteria [24] and giant viruses [25]. Concerning these peculiar association of amoebae with internalised bacteria, it is particularly interesting to compare this relationship with the association of macrophagic cells, a cell type also characterised by the ability to develop pseudopods, with several intracellular pathogenic bacteria such as *Chlamydia* and *Rickettsia* and pathogenic bacteria that develop autophagy mechanisms [26].

## 3. Amoebae and Fish Diseases

As fish are aquatic organisms, it is understandable that fish and amoebae can interact in different ways. The presence of amoeba or amoeboid organisms in association with fish has been described since the beginning of the 20th century. In some cases, these descriptions were limited to the observation or isolation of amoebae from fish, without a clear description of the specific relationship with the fish or even the specific fish tissue. As FLA are ubiquitous organisms in aquatic environments, it is not unlikely that the amoebae ‘isolated’ from fish simply correspond to sampling contamination during the capture or manipulation of the fish. This situation is particularly probable for the identification of amoebae isolated from the fish skin or gills. However, in many descriptions, the relationships between amoeba and the fish are more clear and solid. In this scenario, most of these relationships are based on four different types: ectocommensals, ectoparasites, endocommensals and endoparasites.

As ectocommensals, it is important to stress that particularly gills and also skin constitute a favourable ground for the development of amoebae. Gill and skin mucous can represent a source of nutrients for the amoebae and moreover, the gill and skin microbiome can also be a valuable source of bacterial cells for the feeding activities of the amoebae, in the same way as bacterial biofilms in the environment [27] and even other water storage places [28,29]. All these environments represent a relevant source of nutrients for grazing amoebae. In recent years, the fish intestinal, gill and skin microbiomes are attracting increasing attention in its role in the different epithelial mucosa as the first defence barrier against pathogens. It seems that the mucous surface acts like some kind of ‘no man’s land’ with a delicate and very complex balance between the potentially enhancing substances (mucins, glucose, other carbohydrates, nucleosides, free aminoacids, small peptides) and limiting factors (mostly enzymes such as lysozyme, antibodies, complement factors, antimicrobial peptides (AMPs) such as piscidin, lectins and IFN) present in the mucus composition [30,31]. A similar explanation can be given for endocommensals, as most of the endocommensal amoebae described in fish are found in the digestive tract and only in a few cases are associated with the disease [32]. The mutualistic relationship and relevance of the intestinal microbiota for the digestion and other physiological functions in animals is widely known, so the intestinal tract is also a suitable aqueous microenvironment for the development of endocommensal amoebae.

Amphyzoic species have demonstrated their ability for an endozoic way of life also in aquatic environments. Dyková et al. [33] underlined the pathogenic potential of *Naegleria* sp. for fish since, although most of the *Naegleria* isolates in different freshwater fish species came from clinically healthy fish (mainly from gills), in some cases they were present in internal organs and even in systemic infections [33,34]. *Acanthamoeba* spp. are also able to colonise the organs of freshwater fishes [35,36], and the strains isolated in some cases were closely related to those commonly isolated from cases of human infections, especially *Acanthamoeba* keratitis [36]. Amphyzoic characteristics increase the capacity to survive and increases the complexity of their control, as reinfections are much more likely due to their resilience in the environment and therefore, with limited possibilities for the control. Although these amphyzoic amoebae can cause granulomas in different organs of the host, they are typically isolated from normal organs, like other amoebae genera such as *Protacanthamoeba, Vexillifera* or *Vannella* [37,38,39]. *Protacanthamoeba bohemica*, a closely related species of the genera *Acanthamoeba* and *Balamuthia*, was isolated from the liver of tench (*Tinca tinca*) [38]. *Vannella* spp. are very common in isolates of various organs from freshwater and marine fish, particularly from the gills [39,40,41,42]. Although in some studies, species of *Vannella* were isolated from Atlantic salmon and rainbow trout with lesions in gills [43,44,45], their role in these diseases is unknown and, in general, they are considered non-pathogenic species. *Vexillifera expectata* was isolated from the liver of perch (*Perca fluviatilis*) [37] and more recently three new *Vexillifera* species (*V. bacillipedes*, *V. multispinosa*, *V. fluvialis*, *V. tasmaniana*) have been isolated from the gills and internal organs of freshwater fish [46], without causing apparent lesions. *Thecamoeba* has been isolated from the healthy gills of turbot (*Scophthalmus maximus*) [40] and years ago Sawyer et al. [47] pointed to *T. hoffmani* as the causative agent of gill disease and mortalities in freshwater salmonid fingerlings. *Cochliopodium* spp. are a scale-bearing amoeba that inhabit both marine and freshwater environments [48,49]. *C. minus* was also described from different internal organs and the gills of asymptomatic perch (*Perca fluviatilis*) [50] and the catfish (*Heteropneustes fossilis*), the latter after experimental infection. A similar species to *C. minus* was also isolated from the spleen of gudgeon *Gobio gobio* and the gills of wels catfish (*Silurus glanis*) [40]. *Saccamoeba* sp. was isolated from dead aquarium fish [51] and amoeba isolated from the gills of Atlantic salmon *Salmo salar* was phylogenetically related to *S. limax*, although the ultrastructure did not allow its identity confirmation [52]. *Nolandella* has been isolated from the healthy gills of Atlantic salmon (*Salmo salar)* [43], turbot (*Scophthalmus maximus*) and rainbow trout (*Oncorhynchus mykiss)* [53,54]. Therefore, it does not seem to be a pathogenic genus, however English et al. [41] have also isolated it from symptomatic gills, along with other amoeba species. *Nuclearia pattersoni* was described from the gills of roach (*Rutilus rutilus)* [55].

Among all these different genera of amoeba, probably the most studied species are those that have been isolated or detected on the gills of Atlantic salmon (*Salmo salar*), and that may accompany some type of gill lesions, such as *Acanthamoeba, Flabellula, Heteroamoeba, Vannella, Vexillifera, Mayorella* or *Nolandella* [41,43,56], but their pathogenic role for the moment remains unknown.

Concerning the more specific relationships of pathogenic amoebae, some species have been described as truly pathogenic for fish in the same way as in other vertebrates. However, due to the special characteristics of the fish as 100% aquatic vertebrates, diseases and pathology associated with amoeba can be clearly differentiated in amoebic gill diseases and internal/systemic amoebic diseases. Both present remarkable peculiarities and differences and for this reason are presented in two different sections.

### 3.1. Gill Diseases Caused by Amoebae

Gill diseases associated with amoebae are mainly produced by Amoebozoa, most of them lobosan amoebae, in other words, amoeba with lobosan-shaped (broad and round) pseudopodia. Amoebic gill diseases can be present in marine fish and freshwater fish and although the causative agents can be detected in fish in the wild, disease is only described in fish under aquaculture or captivity conditions. In marine fish, the most widely known disease associated with amoebic organisms is AGD (amoebic gill disease) associated with *Neoparamoeba perurans*, and in freshwater the disease is usually described as NGD (nodular gill disease) with different species of amoebae involved in this complex.

#### 3.1.1. Amoebic Gill Disease (AGD)

AGD is nowadays one of the most relevant diseases in Atlantic salmon (*Salmo salar*) farming in different geographical locations (Norway, Scotland, Ireland, Australia, Chile), with a substantial disease burden including high mortality (sometimes up to 85% in a total production cycle) and a substantial reduction in growth. It is also one of the most relevant gill diseases in this species [57]. The disease and particularly the fish response, as it plays a major role in the pathogenesis and effects of the disease, has recently been extensively reviewed [58]. Although the main impact of the disease is in Atlantic salmon, other salmonids such as coho salmon (*Oncorhynchus kisutch*), chinook salmon (*Oncorhynchus tshawytscha*), rainbow trout (*Oncorhynchus mykiss*) and brown trout (*Salmo trutta*) can be affected. Particularly, rainbow trout can also be affected in freshwater by other amoebic species, but this is usually referred to as nodular gill disease and is presented in the following section. Flatfish species can also be affected and farmed turbot (*Scophthalmus maximus*) is the species where AGD has had a more relevant impact, but the disease and/or the pathogen have been described in other species such as halibut (*Hippoglossus hippoglossus*) [59] and Senegalese sole (*Solea senegalensis*) [60]. Other species where AGD is described are sparidae species [54,61,62], such as sea bream (*Sparus aurata*), sharpsnout seabream (*Diplodus puntazzo*) and black seabream (*Acanthopagrus schlegelii)* and also species such as sea bass (*Dicentrarchus labrax*) [63], ayu (*Plecoglossus altivelis*) [64], rock bream (*Oplegnathus fasciatus*) and grey mullet (*Mugil cephalus*) [62]. However, in these species, the available information about the disease is much more limited than in salmon. A particular case is the infection described in species such as ballan wrasse (*Labrus bergylta*) [65], corkwing wrasse (*Symphodus melops*) and lumpfish (*Cyclopterus lumpus*) [66]. In these species, the development of AGD is probably due to their close contact with potentially affected Atlantic salmon in cages, due to the frequent use of these species as cleaner fish as a system for the biological control of sealice (*Lepeophtheirus salmonis*) in Atlantic salmon farms. All these affected species are also widely reviewed in Oldham et al. [67] and Marcos and Rodgers [58].

**The causative agent** of AGD is the amphyzoic amoebae *Neoparamoeba perurans* (formerly known as *Paramoeba perurans*). Some years ago, when the disease was first described and studied in the USA and Tasmania in different salmonids reared in marine water, the disease was initially associated with the species *Paramoeba pemaquidensis* [68,69]. In the following years, more detailed studies based on morphological and molecular characterisation of *Neoparamoeba* strains isolated from the gills of sea-caged Atlantic salmon but also from marine sediments and the net material of sea-cages were developed [70], stressing the differences between the fish-pathogenic *Paramoeba* species and other non-pathogenic *Paramoeba* free-living species isolated in the gills of affected fish such as *Paramoeba branchiphila* and *P*. *aestuarina*. A few years later, further studies using molecular biology including culture-isolated *P. pemaquidensis* and *P. branchiphila* and also non-cultured, gill-derived (NCGD) amoebae from AGD-affected Atlantic salmon [71] allowed the identification of these NCGD amoebae as the organism responsible for AGD, and the naming of this species as *Neoparamoeba perurans* n. sp. The species name was selected after the Latin verb ‘peruro’ that means ‘burn up, consume, or inflaming’, referring the reaction displayed in AGD, although inflammation is not the main pathological component observed in AGD. The new molecular tools developed from this study were applied in archived and new samples in order to clarify the implication of this new species in the development of AGD lesions [71] and reinforcing its role as the aetiological agent for AGD. Some years later, the genus *Neoparamoeba* was reviewed as a consequence of a study on mortalities on sea urchins associated with amoebae [72] and the name of the genus was moved to *Paramoeba*. However, in 2019 the name of the genus was reverted again to *Neop**aramoeba,* and the former name is considered as a synonym.

**The disease**. AGD is typically a chronic disease, but with a relatively fast development [72], affecting not only gills but also causing a progressive systemic involvement of other organs due to their effects on respiratory, osmoregulatory and circulatory functions. Affected fish become lethargic, with reduced swimming speed, feeding rates and growth and display signs of respiratory distress (increased rate of ventilation).

It is characterised by the development of a progressive hyperplasia of epithelial gill cells in the primary and secondary (lamellae) filaments that leads to an increase of the thickness of the gill epithelia, progressive formation of synechia and lacunae between filaments and finally total obliteration of the interlamellar space [73]. These alterations in gill structure lead to an impairment in gas exchange through the gills and acid-base regulation and consequently in the respiratory function and general metabolism. These changes in the respiratory capacity and in the acid-base homeostasis in the affected fish can also affect the metabolism of other organs such as liver or heart, although there are still discrepancies about the real effects in the gas exchange, oxygen transport and aerobic metabolism in the fish [72].

As AGD also affects the whole structure of the gill epithelium including chloride cells [74], physiological disturbances in the osmoregulation are also involved in the physiopathogenesis of the disease. The gill mucus is also affected and in addition to the increase in mucus production, relevant changes have been detected in the gill mucus in AGD affected fish [75].

**Factors**. High water temperatures (the optimal range for AGD development is 10–18 °C) and also high salinity are considered as two of the main risk factors for the development of the disease and in fact, in some places the outbreaks display a substantial connection with these factors. When seawater salinities drop under 28 per thousand the disease usually diminishes in its intensity. After a first infection, salmon can develop a certain resistance against the disease [76]

Since *N. perurans* is considered an amphyzoic free-living amoeba, the role of the gill microbiota in the ecology of amoebae in this specific microenvironment has recently received attention [77].

**Diagnosis.** AGD diagnosis is normally done through the identification of macroscopic and microscopic lesions in the gills. Macroscopically, AGD is usually characterised by the development of white patches that can develop in the respiratory surface of all four gill arches (Figure 1A). A scoring system based on macroscopic lesions [78] has been recently developed to assess the evolution of the disease. Amoebae can also be seen in gill wet mounts, although sometimes cannot be easily seen as they can remain attached to the gill epithelium or entangled in the proliferative epithelial tissue and lacunae formed in the gills.

Histopathology is usually the most frequently used diagnostic method to complement macroscopic gill assessment, as AGD lesions are quite characteristic (Figure 1B), and allows a more accurate assessment of the gill condition and stage of the disease. The lesions are those previously indicated in the description of the disease, and amoebae can be more easily recognised inside the lacunae (Figure 1C). However, AGD-related lesions are not pathognomonic, as some other chronic and proliferative gill diseases may display a similar histopathologic pattern. Particularly CGD (complex gill disease) is the current term used to describe chronic proliferative and inflammatory problems in the gills of Atlantic salmon [79] and although in many cases AGD and *Paramoeba perurans* can be diagnosed as a single and specific pathological entity, many times AGD can be involved in multifactorial cases of CGD.

As many other non-pathogenic amphyzoic amoebae can be found in gills with lesions, a complementary identification of the presence of *N. perurans* in the gills is recommended to have a robust diagnostic of AGD. This can be done by routinely sampling the gills using cotton swabs and PCR analysis for the detection of *N. perurans* [80].

**Disease management.** AGD management and treatment is usually difficult and complex. Although some oral treatments such as bithionol were tested in the past [81], freshwater and hydrogen peroxide baths are currently the two most frequently used treatments and recently, peracetic acid has also been included in the potential weaponry against this disease [82]. Freshwater baths have been the traditional control system used in Tasmania, but this treatment is highly dependent on the availability of large volumes of freshwater. As good quality freshwater from drinking water networks, rivers, lakes or reservoirs is not always available everywhere, hydrogen peroxide has been considered an acceptable option. Baths are usually performed with the use of tarpaulins in the cages, specific treatment cages already provided with tarpaulins, barges or more recently, well boats, some of them equipped with reverse osmosis systems to supply the ship with freshwater. However, the efficacy of the treatments and the safety of the treated fish are also highly dependent on the environmental conditions, the water quality used in these treatments [83] and also on the fish health condition prior to treatment.

For example, hydrogen peroxide treatment is most efficient at low temperatures [84] and also toxicity increases at high temperatures. This is the reason why AGD treatments should be done as preventive baths during cold seasons and before the onset of the clinical problems at warmer temperatures. This is the most efficient way to keep AGD at bay as it is really difficult to treat and manage once the process is triggered. Regular macroscopic assessments of the aspect of the gills in the different cages or batches using a scoring system is highly recommended and a normal practice in the farms under AGD risk.

Other AGD management strategies include functional feeds, genetic selection and vaccination.

Commercial functional feeds are used to control and reduce the impact of the disease. Most of these feeds are physiology and immunosupportive-based diets formulated to reduce the impact of the disease by increasing immune protection and enhancing the fish recovery.

After some years of research in this field, roe and fry from specific breeds with increased resistance to AGD are nowadays commercially available.

There is a relevant interest in the study of the development of the immune response in AGD mainly to find an efficient protective vaccine, but also to understand the full host response of the salmon immune system against the parasite [58], as the proliferative response triggered by the parasite is itself, one of the problems associated with the disease. Some experimental vaccines were tested in the past and nowadays there are several projects with different vaccine development strategies. However, at this moment there are no available commercial vaccines against AGD.

#### 3.1.2. Nodular Gill Disease (NGD)

A similar pathological condition related to amoebic infection in the gills has also been described in freshwater fish species. This disease is frequently described as nodular gill disease (NGD) but occasionally it has been referred to as proliferative gill disease (PGD) [85]. The use of the name PGD to describe a disease is not recommended as PGD refers to a general epithelial proliferative condition described in a wide number of gill diseases in several fish species associated with different pathogens (e.g., bacterial gill disease (BGD) in salmonids associated with *Flavobacterium*, hamburger gill disease or PGD from *Henneguya* in catfish [86] or complex gill disease (CGD) as a multifactorial syndrome in salmon [79]).

In some ways, NGD can be described as the counterpart of AGD in freshwater, although there are some substantial differences between AGD and NGD that need to be highlighted.

**The causative agent(s).** NGD was described in North America as a specific gill disease affecting rainbow trout [87] usually associated with bacterial gill diseases and in this paper a large number of protozoa were described associated with these lesions. Sometime later, the disease was associated with the presence of so-called ‘A cells’ suggesting similarity of these cells to amoebae [88]. These ‘A-cells’ were lately recognised as amoebae causing similar lesions in rainbow trout and also in arctic charr (*Salvelinus alpinus*) and tentatively identified as *Cochliopodium* sp. [89]. Since then, several reports of NGD mainly in farmed rainbow trout and also brown trout (*Salmo trutta*) have been described in North America and Europe [90] and also in farmed chinook salmon (*Oncorhynchus tshawytscha*) [91].

The first description of amoebae in farmed salmonid gills was from Sawyer et al. [47]. However, there are some other previous descriptions of amoebic organisms in salmonid fingerling gills, but not associated to NGD or disease. These amoeboid organisms were initially characterised as *Thecamoeba hoffmani* and *Schyzamoeba salmonis*. However, the accurate identification of the species was based on general microscopic features and not on more suitable methodologies. Some years later, Dyková et al. [44] published a study based on amoebae isolated in rainbow trout (*Oncorhynchus mykiss)* farms in Germany, identifying different strains of *Acanthamoeba*, *Hartmannella*, *Naegleria*, *Protacanthamoeba* and *Vannella* from the gills with clear NGD lesions. However, in this study it was not possible to identify which of them were the main organisms responsible for the lesions observed. The authors suggested *Acanthamoeba*, *Hartmannella* and *Protacanthamoeba* as the species with highest pathogenic potential. Again, as in marine amoebae, the main challenge was the identification of the potentially pathogenic amoebae from the relatively high diversity of free-living amoebae that can be present in the gills. Later studies [45] also identified another species, *Roghostoma minus* highly related with NGD infections, highlighting the heterogeneous diversity of amoebic species potentially involved in the NGD development in freshwater, in contrast to the scenario in seawater, where AGD is mainly related with one species.

NGD was and still is a very common disease in rainbow trout farms in Europe [85], although its relevance and burden has not been properly addressed until recently. Recently, some studies on NGD outbreaks in Italy [47,90,91,92] highlighted the re-emergence of NGD as a relevant disease in trout farming.

**The disease.** NGD symptomatology in rainbow and brown trout is similar to the symptomatology in AGD-affected salmons: fish are less active, display respiratory distress, changes in swimming behaviour and reduced growth. Mortality is usually noticed some weeks after the first symptoms appear. The external aspect of the lesions in the gills is also characterised by the presence of whitish areas in the gill filaments and increased presence of mucus. Microscopic histopathological lesions are characterised by a particularly intense epithelial hyperplasia leading to partial or complete lamellar fusion and even, fusion between adjacent filaments. As a point of difference with AGD, it is important to highlight that the presence of inflammatory infiltrate has been described associated with the lesions [47,90,91,92] although in some cases, very weak or no inflammation is also seen (personal observations). In NGD, the distal region of the filaments is usually the most affected part of the gills (in contrast to AGD) and sometimes filaments display a club or maze aspect. Here, the epithelial hyperplasia is particularly intense and sometimes displays spongiosis. In the most advanced cases amoebae can be seen as flattened organisms attached to the surface of these proliferative areas or in the middle of the lacunae formed between the fused lamellae (Figure 1D,E).

**Factors.** The epidemiology and factors affecting NGD development in freshwater salmonids is much less understood than in seawater AGD so the role of water characteristics is not known. NGD outbreaks are usually associated with other mixed pathological conditions and infections (other protozoan parasites, monogeneans, BGD by *Flavobacterium*, *Saprolegnia*) that makes the diagnosis and the prognosis more complex and difficult.

**Diagnosis.** The diagnosis is usually done by combined macroscopic and histopathological lesions. The identification of amoebae present in this case has a scarce clinical relevance as many different species of amoeba can be found [44,45]. However, from the scientific point of view it is necessary to have a major insight in the amoebae involved in the NGD cases and particularly in the specific role of the species in the pathogenesis of the disease.

**Disease management.** Very little information is available about treatments for NGD. General treatments for ectoparasites (formalin) can be effective but no specific data on that is available. No effect was found in 10 ppm chloramine T baths (1 h), although this treatment was mainly directed to treat a BGD co-infection [91]. Surprisingly, no data on the use of salt baths are described in the literature. As freshwater is one of the most efficient treatments to control marine amoebae in AGD, the potential effect of salt baths cannot be ruled out simply using an osmotic shock to kill and remove the amoeba. In fact, salt (sodium chloride) is one of the best, easiest, cheapest and sometimes forgotten options used to treat certain ectoparasites and reduce osmotic stress in freshwater fish. Repeated short treatments (10–20 min) with 10–30 g/L salt are usually recommended to control ectoparasites (ciliates, monogeneans) in aquaculture, so its use to control freshwater NGD is highly promising.

### 3.2. Systemic Diseases Caused by Amoebae

Systemic amoebic infections in fish have been known for a long time, but they have been reported causing significant lesions in internal organs only occasionally. Systemic granulomatous infection due to an amoebae-like organism was described 40 years ago in goldfish (*Carassius auratus*) [93,94,95,96]. Systemic infection by amoebae-like organisms were also occasionally described in other fish species such as dwarf (*Colisa lalia*) [97], European wels catfish (*Silurus glanis*) [98] or pompano (*Trachinotus falcatus*) [61]. More recently, a similar pathological condition was described in Senegalese sole (*Solea senegalensis)* [99] and tench (*Tinca tinca)* [100]. In both studies, due to the absence of mitochondria in the amoebae-like organisms detected, Archamoebae were suggested as the aetiological agent of the infection. Amoebae belonging to the class Archamoebae, contrary to free-living amphizoic amoebae, are characterised by the lack of mitochondria [15,101,102,103], probably related to their adaptation to parasitism. Among these parasitic amoebae species, the genus *Entamoeba,* as stressed before, is the most important one because it parasitises several vertebrate species (e.g., [15,100,101,102]), including fish [94]. *Entamoeba salpae* [104], *E. gadi* [105], and *E. molae* [106] were described from marine fish species, whereas *E. pimelodi* [107], *E. ctenopharyngodoni* [108] and *E. chiangraiensis* [109] have been described from freshwater fish. Excluding *Entamoeba*, most members of Archamoeba are not well studied, probably due to the relatively minor clinical importance and the difficulties associated with their laboratory cultivation [110]. Fish are not an exception, and the majority of mentions of these amoebae refer to specific findings or isolation in a specific fish species. However, the systemic granulomatosis in Senegalese sole (*Solea senegalensis)*, has sparked considerable interest due to the economic losses it causes in the fish farming industry and has been investigated in more detail.

#### Amoebic Granulomatous Disease in Cultured Senegalese Sole

The amoebic granulomatous disease of farmed Senegalese sole (*Solea senegalensis*) in the Spanish Atlantic coast was first described in 2010 [99]. Affected fish presented protuberances or lumps in the muscle, often noticeable at the skin surface (Figure 2A).

The disease does not cause a high mortality but the prevalence in the farms can be very high [111], causing severe economic problems as the affected fish are unmarketable due to their aspect.

Since then, this pathological condition has discontinuously affected different farms on the European Atlantic coast.

Unicellular organisms were initially found in histopathological studies closely associated with the granulomatous lesions, strongly suggesting a parasitic origin of the disease. The differential diagnosis from other parasitic problems included different groups of protozoans such as flagellates, apicomplexa, microspora, mesomycetozoa or amoebae [99]. Finally, ultrastructural features of the organisms clearly pointed to amoeba and molecular studies allowed the identification of *Endolimax piscium* as the causative agent [112].

**The causative agent.** The archamoebic species *Endolimax piscium* is a species phylogenetically related to the entamoebids *E. nana* and *Iodamoeba* spp. [112], which are enteric commensals and parasitic species in humans and other animals. Entamoebids usually have two stages in their life cycle [3,110]: a trophozoite stage (vegetative and infectious form, which is the responsible for the disease) and a cyst (a form of resistance, which is the responsible for transmission). Up to now, the only stage characterised at the moment for *E. piscium* is the trophozoite. The trophozoite of *E. piscium* is 2 to 5 μm in diameter and contains one vesicular nucleus with a large and central round nucleolus. Endoplasmatic reticulum, dictiosomes and a variety of structures such as some single-membrane bound vesicles, putative digestive vacuoles containing particulate material, myelinic figures or products of lysosomal action are also described in the cytoplasm [99,112]. Like other members of Archamoebae, they are amitochondriates, but they contain double-membrane-bounded, electron-dense organelles with no apparent cristae resembling mitosomes usually being observed.

**The disease.** Macroscopic lesions in symptomatic fish consist of nodules with an abscess-like aspect located in different tissues, especially in muscular tissue but also in the liver, digestive tract, gonads, heart and kidney. These nodules are clearly distinguishable from the rest of the tissues and usually present a soft and liquefied consistency (Figure 2B).

Histologically, the nodules correspond to chronic granulomatous inflammatory lesions with a large core of homogeneous necrotic tissue surrounded by fibroblasts and macrophages (Figure 2C). Amoebae appear as ring-like layers surrounding the periphery of the lesions, between the external inflammatory reaction and the necrotic tissue, free or within the cytoplasm of macrophages [99]. In some cases, extensive necrosis and diffuse inflammatory areas are also found, especially in muscle and the liver.

In addition to the lesions, *E. piscium* is also frequent within the intestinal epithelium of fish without apparent lesions (Figure 2D). One of the most important issues of this disease is that *E. piscium* can be present in the intestine of both symptomatic (i.e., displaying granulomatous lesions) and asymptomatic fish [99,113,114], which points to an endocommensal behaviour of this species and it is considered the initial stage in the development of the disease [113]. In moderate to high parasite intensity, amoebae can reach submucosa [113], which indicates a proliferation of the parasite and the progression to other organs, with a similar behaviour to *E. histolytica*. An haematogenous dissemination is the most accepted route of liver invasion and other organs in the case of *E. histolytica* trophozoites [115,116]. Although in the case of *E. piscium* this mechanism is still unknown, the connective tissue (probably facilitated by host phagocytic cells) may play a significant role and an haematic route should not be disregarded, especially in advanced infections with high levels of parasitemia [113].

The amoebic granulomatous disease does not cause important mortalities in Senegalese sole (*Solea senegalensis);* actually it is a long-term chronic disease. The affected fish can survive for a long time with lesions. The detrimental effects in fish, such as reduced growth performance and higher susceptibility to other diseases, are evident, and the extensive lesions make the fish unmarketable after harvesting.

**Factors.** The triggering factors or specific conditions for the development of overt clinical infections, that facilitate the transition from an endocommensal to an aggressive and systemic parasite, are still unknown.

Temperature has been suggested as an important factor [99,114], but the appearance of symptomatic fish at any time of the year, even in the cold months of January and February, seems to rule it out.

Many amoeba outbreaks can be favoured by immunodepression or suboptimal environmental conditions [98,117]. Brain lesions in humans caused by *Acanthamoeba* species, *Balamuthia* or even *Entamoeba*, among others, are usually seen in individuals suffering from other chronic diseases, having compromised immune systems, or during outbreaks of bacterial infections [118]. In this sense, Constenla et al. [114] pointed to *Tenacibaculum* infection as one of the important factors that led to positive cases of *E. piscium* in fish muscle during an experimental infection.

Water recirculation conditions, as in the case of the pre-fattening and fattening systems of sole, could be favouring the survival and permanence of these stages in the rearing system, external to the host, facilitating a proliferation of the parasite among the fish particularly in intensive farming condition facilities.

**Diagnosis.** Histopathological observation combined with molecular methods (ISH, PCR and qPCR) are the diagnostic methods currently available for the diagnosis of this disease. The histopathological observation is limited by the minute size and cryptic nature of *E. piscium*, but useful if parasites are associated with the typical granulomatous responses [113]. Diagnosis of subclinical infections by histopathological observation usually is not enough for the confirmation of the diagnostic and only achievable by an in-depth analysis by an experienced pathologist. However, confirmation of the diagnosis can be obtained with high predictive values and high accuracy (>99%) if combined with molecular hybridisation techniques [113]. The ISH efficiently detects *E. piscium* trophozoites present in different tissues, allowing the identification of the parasite even when found in small numbers or non-associated to lesions (Figure 1E) and therefore, is the reference confirmatory method, particularly in asymptomatic fish.

Probes of PCR and qPCR are also available and offer a rapid diagnostic for a high number of samples, but the sampling for these methods is very important so as not to reduce their sensitivity. For instance, the sensitivity of PCR methods decreases in intestine samples with low parasite intensities, probably due to the patchy distribution of *E. piscium* within this tissue, but this sensitivity clearly increases if homogenates of all or a large part of the intestine are used instead of small pieces [113].

**Disease management.** To date, no treatments have been described to control the disease, and therefore prophylaxis appears as the only option for its management. In this context, it is essential to detect not only the parasite in carrier fish [113], but also stages outside the host that may be retained at the bottom of the tanks particularly in recirculation systems. It is important to keep in mind that *Endolimax nana* and entamoebid species commonly include parasites with two stages in their life cycle: a trophozoite and a cyst [110]. The latter is the resistant stage, which allows the parasite to remain in the environment after leaving the host and until reaching new hosts to continue their life cycle. Although no cyst stages have ever been identified in *E. piscium*, positive results by qPCR were obtained from water samples after filtration during an experimental infection [114]. The potential encysting capacity of this amoeba is an important handicap for fish farm prophylaxis because amoeba cysts can be very resistant to temperature, desiccation and disinfection treatments [119]. The same experimental infection demonstrated that *E. piscium* can be horizontally transmitted to healthy fish by cohabitation with no direct contact between fish, only contaminated water [114]. In addition, in intensive culture conditions, where bottom-dwelling flatfish are reared at high density, the faecal−oral contagion route is also likely to occur readily in intestinal parasites.

Early detection of the parasite in the farm should be considered a priority for the management of this disease, as the muscular lesions developed can prevent the commercialisation of the fish after an important investment in their feeding and maintenance.

In most cases, only after a complete and thorough sanitary break and a careful assessment of the absence of the parasite in the new fry or juvenile batches was it possible to eliminate the disease from the facilities. The use of flow-through systems and the reinforcement of hygienic and disinfection technology and husbandry procedures within the recirculation system such as implementation of powerful UV systems, post-filtration disinfection with ozone or similar systems and periodical cleaning of tank bottoms and surfaces can also reduce the risk of the disease in the systems.

## Figures and Tables

**Figure 1 animals-11-00991-f001:**
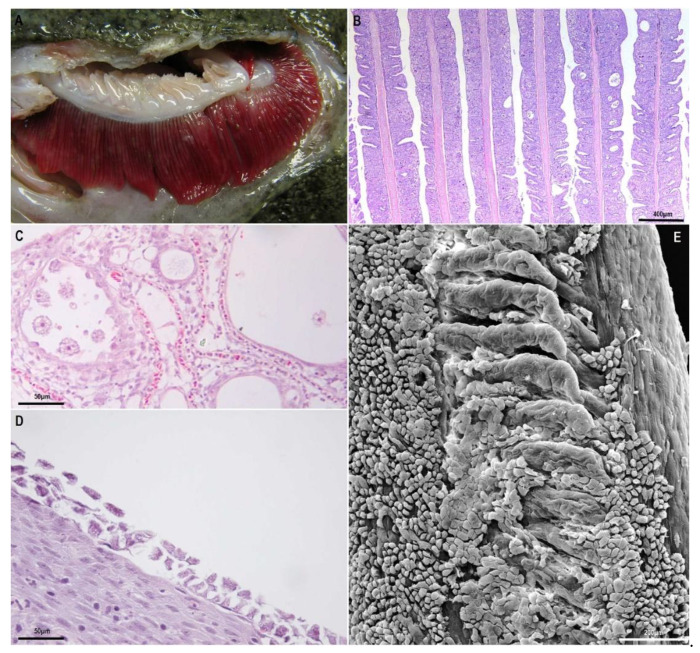
Gills affected by amoebic gill disease (AGD) and nodular gill disease (NGD): (**A**), gill filaments of turbot affected by amoebic gill disease. Note the typical white multifocal lesions. (**B**–**D**), Paraffin-embedded histological sections: (**B**), extensive hyperplasia of epithelial gill cells in Atlantic salmon, with synechiae and lacunae between filaments; (**C**), detail of amoebae within lacunae between gill filaments in AGD in Atlantic salmon; (**D**), flattened amoebae attached to the surface of proliferative gill lesions of NGD in rainbow trout; (**E**), transmission electron micrographs of gill with amoebae attached to the epithelium in NGD in rainbow trout.

**Figure 2 animals-11-00991-f002:**
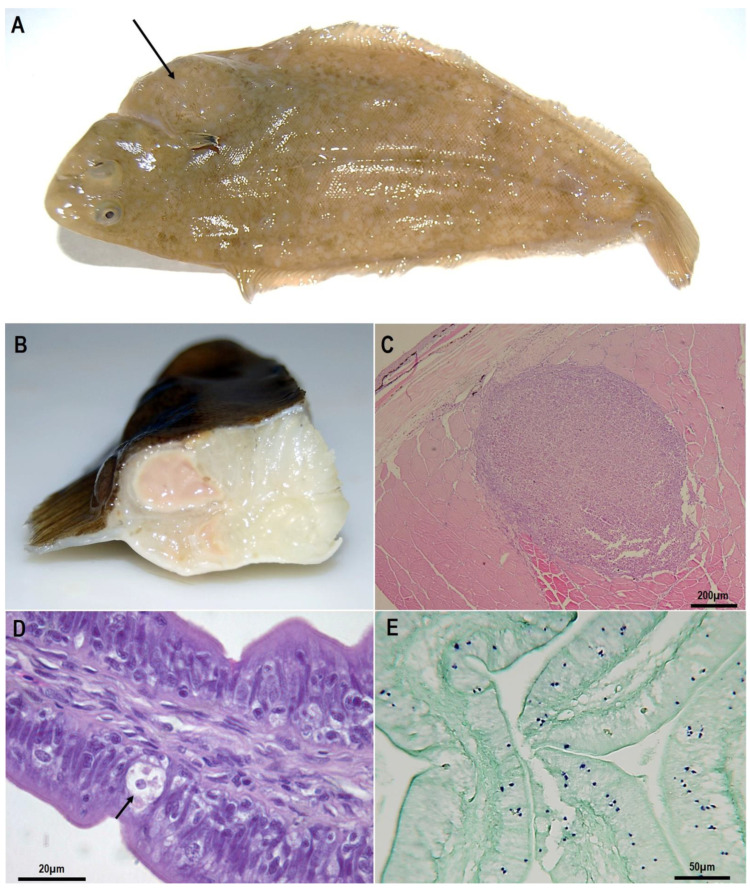
Senegalese soles clinically infected by *Endolimax piscium*: (**A**), conspicuous lump from the abdominal cavity, and noticeable at the skin surface (arrow); (**B**), nodule in muscle with an abscess-like aspect with a soft and liquefied consistency. (**C**–**E**), paraffin-embedded histological sections: (**C**), skeletal muscle showing the typical granulomatous inflammatory reaction to *E. piscium*; (**D**), *E. piscium* cells within the intestinal epithelium (arrow); (**E**), in situ hybridisation of histological sections where parasite cells can be recognised stained in purple.

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
