# Peer review of "Diseases Caused by Amoebae in Fish: An Overview"

_animals, 2021, doi:10.3390/ani11040991_

Round 1
Reviewer 1 Report
The manuscript is well structured and the review feels complete and with up-to-date information. The use of the English language needs some revision. Some comments and suggestions are included in the attached file.

Author Response
Thanks to the reviewer for the extensive constructive feedback, which has led to improve the manuscript. All suggestions have been incorporated or changed in the ms.
Reviewer 2 Report
This careful review about diseases caused by amoeba provides detailed and joint information on the current state of knowledge of these agents, so it should be published.
However, the sections and subsections of the review would be clearer if, for example, a numbered list were used, since sometimes the titles appear in uppercase, others in uppercase and lowercase, and it is not well understood which section belongs to a previous section.
I have also detected multiple small typographical errors, incorrect use of some words or small phrases that are not understood correctly, in such a way that it should be fixed before publication.
Here are my suggestions to improve the manuscript:
- "do not harm fish" line 13.
- Fix extra spaces between words or characters: lines 17, 19, 26, 43, 59 (lacks a space in "E. coli" and there an extra space in "as the most"), 67, 70, 75, 84, 134, 143, 148, 153, 160, 163, 176, 207, 230, 231, 238, 265, 267, 289, 313, 316, 327, 332, 344, 351, 365, 368, 375, 390, 392, 398, 412, 416, 426, 460, 462, 475, 484, 550, 562, 567, 677, 707, 788, 830.
- Avoid capitalized: supergroups (40), granulomatous amoebic encephalitis (64), amoebic keratitis (64), primary amoebic meningoencephalitis (65), amoebic gill disease (188), nodular gill disease (189, 340), complex gill disease (293), bacterial gill diseases (350), amoebae (358), archamoebic (470), entamoebids (470).
- Line 84: it is strange to read "non-human primates: dogs...", better a comma than a colon.
- In line 45 the authors define the abbreviation FLA, but it is used just twice in the text and not throughout the manuscript.
- Use italics: Chlamydia (100), Rickettsia (100), Acanthamoeba (140, 518), P. (227), Neop (239), N. (299, 301), Balamuthia (518), Entamoeba (519) and in the scientific names in the bibliography (lines 594, 598, 601, 602, 603, 605, 619, 625, 629, 642, 652, 655, 665, 666, 668, 674, 680, 681, 684, 687, 689, 692, 710, 711, 716, 717, 723, 726, 727, 729, 730, 732, 733, 737, 738, 745, 746, 748, 749, 751, 755, 758, 761, 767, 770, 775, 779, 782, 791, 797, 801, 804, 807, 809, 814, 823, 828, 836, 837, 841, 842, 843, 852, 854, 857, 858, 860, 861, 864, 867, 868, 871, 872, 878, 880).
- The abreviation of the 20th century is c. XX? I am not sure, but I have not seen this before (107).
- INF? Do authors mean IFN? (128).
- Check the word endocommensal throughout the text. It sometimes appears as "endocomensal" (129, 133, 513...).
- "sp." no italics, lacks a period (136, 354). "spp." lacks a period (156). "or" no italics (170).
- Check the citation style in line 154, 157, 353, 358, 363.
- In some parts of the text the authors mention the common name of fish followed by its scientific name. In other parts the scientific name appears in parentheses, and in others there is not a scientific name. It should be unified.
- "archived" instead of "archival"? (234).
- "Diagnosis" instead of "diagnostic" (270, 271, 402, 403, 528, 540).
- Line 277 lacks a period.
- Check the font in the figure caption (280-282, 450-455).
- The abbreviation PGD in line 341 should be explained the fisrt time it appears in the text.
- "water characteristics" instead of "water temperature and other water characteristics" (398-399).
- "the diagnosis and the prognosis" (401).
- "amoeba-like" or "amoebic-like"? Are both correct? May be used just one of the forms? (424-425).
- "the economic losses it causes" instead of "the economic causes it has caused and causes" (441-442).
- "citoplasm" (491).
- "Senegalese" (506).
- "condition" (526).
- Check the style in the bibliography. The use of capital letters, the periods in the authors, the years in bold, the use of italics... Are all the references complete? i.e. 51, 52, 53?
Author Response
Response to REVIEWER 2
This careful review about diseases caused by amoeba provides detailed and joint information on the current state of knowledge of these agents, so it should be published.
However, the sections and subsections of the review would be clearer if, for example, a numbered list were used, since sometimes the titles appear in uppercase, others in uppercase and lowercase, and it is not well understood which section belongs to a previous section.
I have also detected multiple small typographical errors, incorrect use of some words or small phrases that are not understood correctly, in such a way that it should be fixed before publication.
Authors: Thanks to the reviewer for the extensive and constructive feedback, which has led to improve the manuscript.
Avoid capitalized: supergroups (40), granulomatous amoebic encephalitis (64), amoebic keratitis (64), primary amoebic meningoencephalitis (65), amoebic gill disease (188), nodular gill disease (189, 340), complex gill disease (293), bacterial gill diseases (350), amoebae (358), archamoebic (470), entamoebids (470).
Authors: In our opinion supergroups should be capitalized, but the rest of the words have been corrected in the ms
In line 45 the authors define the abbreviation FLA, but it is used just twice in the text and not throughout the manuscript.
Authors: We do not quite understand what the reviewer is referring to, but FLA is precisely in the introduction to indicate that this concept exists, and some amoebas belong to this ecological concept. It is not necessary to repeat this concept in the text (outside the introduction) since the paper does not focus on free-living amoebae.
Authors: All other suggestions have been incorporated or changed in the ms.